# Debiasing, calibrating, and improving Semi-supervised Learning performance via simple Ensemble Projector

## Abstract

Recent studies on semi-supervised learning (SSL) have achieved great success. Despite their promising performance, current state-of-the-art methods tend toward increasingly complex designs at the cost of introducing more network components and additional training procedures. In this paper, we propose a simple method named Ensemble Projectors Aided for Semi-supervised Learning (EPASS), which focuses mainly on improving the learned embeddings to boost the performance of the existing contrastive joint-training semi-supervised learning frameworks. Unlike standard methods, where the learned embeddings from one projector are stored in memory banks to be used with contrastive learning, EPASS stores the ensemble embeddings from multiple projectors in memory banks. As a result, EPASS improves generalization, strengthens feature representation, and boosts performance. For instance, EPASS improves strong baselines for semi-supervised learning by 39.47%/31.39%/24.70% top-1 error rate, while using only 100k/1%/10% of labeled data for SimMatch, and achieves 40.24%/32.64%/25.90% top-1 error rate for CoMatch on the ImageNet dataset. These improvements are consistent across methods, network architectures, and datasets, proving the general effectiveness of the proposed methods.

## 1 Introduction

Deep learning has shown remarkable success in a variety of visual tasks such as image classification He et al. (2016), speech recognition Amodei et al. (2016), and natural language processing Socher et al. (2012). This success benefits from the availability of large-scale annotated datasets Hestness et al. (2017); Jozefowicz et al. (2016); Mahajan et al. (2018); Radford et al. (2019); Raffel et al. (2020). Large amounts of annotations are expensive or time-consuming in real-world domains such as medical imaging, banking, and finance. Learning without annotations or with a small number of annotations has become an essential problem in computer vision, as demonstrated by Zhai et al. (2019); Chen et al. (2020a;c;b); Grill et al. (2020); He et al. (2020); Laine & Aila (2017); Lee et al. (2013); Sohn et al. (2020); Li et al. (2021); Zheng et al. (2022); Berthelot et al. (2019; 2020); Tarvainen & Valpola (2017); Xie et al. (2020b).

Contrastive self-supervised learning (CSL) is based on instance discrimination, which attracts positive samples while repelling negative ones to learn the representation He et al. (2020); Wu et al. (2018); Chen et al. (2020a). Inspired by CSL, contrastive joint-training SSL methods such as CoMatch Li et al. (2021) and SimMatch Zheng et al. (2022) leverage the idea of a memory bank and momentum encoder from MoCo He et al. (2020) to support representational learning. In the current mainstream contrastive joint-training SSL methods, a multi-layer perceptron (MLP) is added after the encoder to obtain a low-dimensional embedding. Training loss and accuracy evaluation are both performed on this embedding. The previously learned embeddings from a low-dimensional projector are stored in a memory bank. These embeddings are later used in the contrastive learning phase to aid the learning process and improve the exponential moving average (EMA) teacher Tarvainen & Valpola (2017). Although previous approaches demonstrate their novelty with state-of-the-art benchmarks across many datasets, there are still concerns that need to be considered. For instance, conventional methods such as CoMatch Li et al. (2021) and SimMatch Zheng et al. (2022) are based on the assumption that **the learned embeddings are correct, regardless of confirmation bias**. This

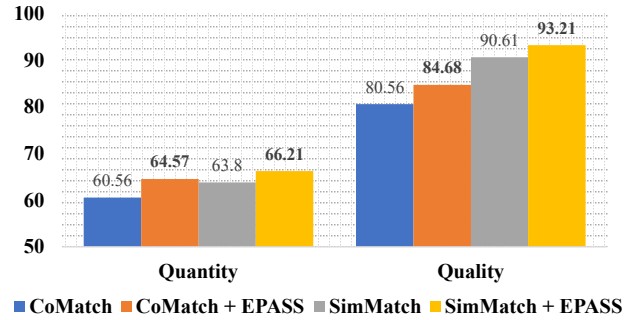

Figure 1: Quantity vs quality of pseudo-labels on ImageNet 10% with and without EPASS.

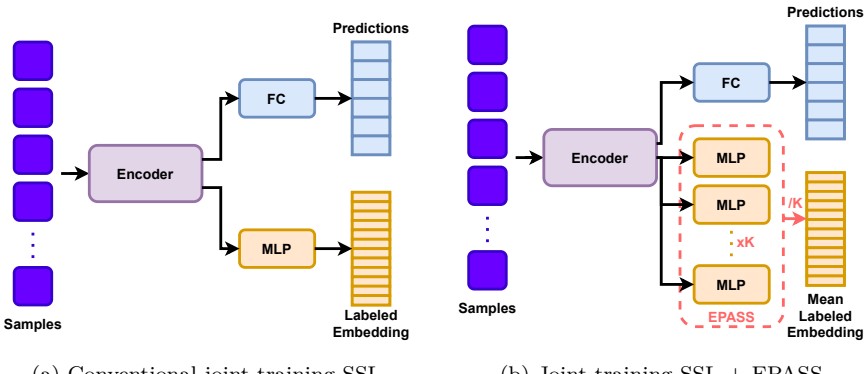

(a) Conventional joint-training SSL          (b) Joint-training SSL + EPASS

Figure 2: Training phase for contrastive joint-training SSL without/with the proposed EPASS. 2a represents the conventional training phase without EPASS Li et al. (2021); Zheng et al. (2022). Unlike 2a, in 2b, instead of using **only one projector** to learn the embeddings, EPASS uses **multiple projectors** to ensemble the embeddings, which is less biased and more generalized.

assumption is directly adopted from CSL; however, in a joint-training scheme, the easy-to-learn representation could easily dominate the hard-to-learn representation, leading to biased distributions and embeddings. This would become even worse when confirmation bias happens and the embeddings are driven away by the incorrect pseudo-labels. As a result, the embeddings stored in the memory bank are also affected, causing the confirmation bias issue and the erroneous EMA teacher.

The confirmation bias could be seen in Figure 1, where CoMatch only has **80.56%** correct pseudo-labels and SimMatch has **90.61%** correctness for pseudo-labels. When the embedding bias happens at the instance level and the confirmation bias happens at the semantic level, they degrade the performance of the EMA teacher. As a result, the well-learned embeddings at the instance level could be driven away by the confirmation bias at the semantic level during backward propagation, and vice versa.

To address these limitations, we propose **E**nsemble **P**rojectors **A**ided for **S**emi-**s**upervised Learning (EPASS), a plug-and-play module to strengthen the EMA teacher as well as to improve the generalization of the learned embeddings, as illustrated in Figure 2. Adding a projector helps mitigate the overfitting problem, and the generated features are more distinguishable for classification Li et al. (2021); Zheng et al. (2022). Chen et al. (2022b) proves the strengths of ensemble projectors in teacher-student frameworks via knowledge distillation. Therefore, we leverage those strengths with SSL, especially contrastive joint-training frameworks. Although there has been study about ensemble for SSL Chen et al. (2022a), they only discover it in the classification head, thus resulting in a large number of parameter overheads as shown in Table 1. Unlike Chen et al. (2022a), we specifically enrich the learned embeddings from the model by employing multiple projectors rather than

| Method | WRN-28-2 | WRN-28-8 |
|---|---|---|
| Original | 1.4 M | 23.4 M |
| Chen et al. (2022a) | 3.7 M (↑ 2.3) | 19.9 M (*, ↓ 3.5) |
| CoMatch Li et al. (2021) | 1.5 M | 23.71 M |
| SimMatch Zheng et al. (2022) | 1.5 M | 23.74 M |
| **CoMatch + EPASS (3 projs)** | 1.54 M (↑ 0.04) | 24.30 M (↑ 0.59) |
| **SimMatch + EPASS (3 projs)** | 1.56 M (↑ 0.06) | 24.39 M (↑ 0.65) |

Table 1: Comparison with multi-head co-training. '*' indicates different architecture as Chen et al. (2022a) modified the number of channels of the final block from 512 to 256.

only one, as it is common in conventional methods. Using ensemble projectors in contrastive learning, where multiple projectors are used instead of a single one, may improve the performance and robustness of the learned representations. By using multiple projectors, the model can learn different feature representations from different perspectives, which can be combined to produce more informative representations of the data. Additionally, using ensemble projectors can help to improve the generalization performance of the model, by reducing the risk of overfitting to the specific characteristics of a single projector.

Using ensemble projectors can also increase the robustness of the model against variations in the data distribution, as the multiple projectors can learn different features that are less sensitive to changes in the data distribution. This can be especially useful in situations where the data distribution is not well-defined or changes over time. Therefore, the embeddings of the model would be the ensemble ones, which are less biased and more robust than conventional methods. Our comprehensive results show that such a simple ensemble design brings a sweet spot between model performance and efficiency.

By incorporating the ensemble projectors in a contrastive-based SSL fashion, the proposed EPASS makes better use of embeddings to aid contrastive learning as well as to improve the classification performance simultaneously. In addition, ensemble multiple projectors introduce a relatively smaller number of parameters compared with ensemble multiple classification heads. Extensive experiments justify the effectiveness of EPASS, which produces a less biased feature space. Specifically, EPASS achieves a state-of-the-art performance with **39.47%/31.39%/24.70%** top-1 error rate, while using only 100k/1%/10% of labeled data for SimMatch; and achieves **40.24%/32.64%/25.90%** top-1 error rate for CoMatch on ImageNet dataset.

The contributions of this paper are summarized as follows:

- We hypothesize that the conventional contrastive joint-training SSL frameworks are sub-optimal since the multi-objective learning could harm the learned embeddings when confirmation bias occurs.

- We propose EPASS, a simple plug-and-play module that improves a generalization of the model by imposing the ensemble of multiple projectors, which encourages the model to produce less biased embeddings.

- To the best of our knowledge, this is the first work to enhance the performance of contrastive joint-training SSL methods by considering the embedding bias.

- Extensive experiments on many benchmark datasets demonstrate that EPASS consistently improves the performance of contrastive joint-training methods.

## 2 Related Work

### 2.1 Semi-supervised Learning

Semi-supervised learning is an essential method to leverage a large amount of unlabeled data to enhance the training process. Pseudo-label Lee et al. (2013) is the pioneer of nowadays popular methods, including self-

training-based or consistency-based SSL approaches. In the pseudo-label-based methods, the model is first trained on a small amount of labeled data. Then, the model is used to make predictions for unlabeled data. The unlabeled data and their corresponding pseudo-labels are then used to train the model simultaneously with labeled data, forming the self-training-based methods Lee et al. (2013); Arazo et al. (2020); McLachlan (1975); Tarvainen & Valpola (2017); Zhang et al. (2019); Bachman et al. (2014); Xie et al. (2020b). Consistency-based methods Sohn et al. (2020); Zhang et al. (2021); Berthelot et al. (2019; 2020); Miyato et al. (2019); Zheng et al. (2022); Li et al. (2021) use a high threshold to determine the reliable predictions from weakly augmented samples. Then, they will be used as pseudo-labels for strongly augmented examples, and the low-confidence predictions will be discarded. However, those approaches suffer from confirmation bias Arazo et al. (2020) since they overfit the incorrect pseudo-labels during training. Moreover, methods using the high threshold to filter noisy data only use a small amount of unlabeled data during training, and when the model suffers from confirmation bias, it leads to the *Matthew effect*.

Sohn et al. (2020) introduces a hybrid method named FixMatch, which combines pseudo-labeling with a consistency regularization method. By using a high threshold to filter out noisy pseudo-labels, FixMatch lets the model learn from only confident predictions, thus improving its performance. FlexMatch Zhang et al. (2021) introduces a Curriculum Pseudo Labeling (CPL) method based on the Curriculum Learning (CL) Bengio et al. (2009). CPL configures a dynamic threshold for each class after each iteration, thus letting the model learn better for either hard-to-learn or easy-to-learn classes.

## 2.2 Contrastive joint-training SSL

Li et al. (2021) proposes CoMatch, which combines two contrastive representations on unlabeled data. However, CoMatch is extremely sensitive to the hyperparameter setting. Especially during training, CoMatch requires a large memory bank to store the embedded features. Recently, Zheng et al. (2022) published work that takes semantic similarity and instance similarity into account during training. It shows that forcing consistency at both the semantic level and the instance level can bring an improvement, thus achieving state-of-the-art benchmarks. Along this line of work, Yang et al. (2022); Zhao et al. (2022) also leverage the benefit of Class-aware Contrastive loss to the training process of SSL.

Previous methods might fail to provide the correct embeddings due to confirmation bias. Conventionally, confirmation bias does not exist in CSL; however, it occurs in contrastive joint-training SSL by the use of a threshold. It leads to the degradation of the classifier and the projector, thus providing incorrect predictions and embeddings. Our EPASS exploits the ensemble strategy for multiple projectors, imposing consistency and improving generalization for the learned embeddings, thus enhancing the correctness of model predictions.

# 3 Method

## 3.1 Preliminaries

We first define notations used in the following sections. For semi-supervised image classification problem, let $\mathcal{X} = \{(x_b, y_b) : b \in (1, \ldots, B)\}$ be a batch of $B$ labeled examples, where $x_b$ is training examples and $y_b$ is one-hot labels, and $\mathcal{U} = \{u_b : b \in (1, \ldots, \mu B)\}$ be a batch of $\mu B$ unlabeled examples where $\mu$ is a hyperparameter determining the relative sizes of $\mathcal{X}$ and $\mathcal{U}$. For labeled samples, we apply weak augmentation ($\mathcal{A}_w$) to obtain the weakly augmented samples. Then, an encoder $f(\cdot)$ and a fully-connected classifier $h(\cdot)$ are applied to get the distribution over classes as $p(y \mid x) = h(f(x))$. The supervised cross-entropy loss for labeled samples is defined as:

$$\mathcal{L}_s = \frac{1}{B} \sum_{b=1}^{B} \mathcal{H}(y_b, p_b) \tag{1}$$

where $\mathcal{H}$ is a standard cross-entropy loss function.

Conventionally, CoMatch and SimMatch apply a weak ($\mathcal{A}_w$) and strong ($\mathcal{A}_s$) augmentation on unlabeled samples, then use the trained encoder and fully-connected classifier to get the predictions as $p_b^w = p(y \mid \mathcal{A}_w(u_b))$

and $p_b^s = p\left(y \mid \mathcal{A}_s\left(u_b\right)\right)$. Following CoMatch Li et al. (2021) and SimMatch Zheng et al. (2022), the predictions that surpassing confidence threshold $\tau$ would be directly used as pseudo-labels to compute the unsupervised classification loss as:

$$\mathcal{L}_u = \frac{1}{\mu B} \sum_{b=1}^{\mu B} \mathbb{1}\left(\max\left(\hat{p}_b^w\right) \geq \tau\right) \mathrm{H}\left(\hat{p}_b^w, p_b^s\right) \tag{2}$$

where $\hat{p}_b^w = DA\left(p_b^w\right)$ is the pseudo-label for input $\mathcal{A}_w\left(u_b\right)$ and $DA$ is the distribution alignment strategy Li et al. (2021); Zheng et al. (2022) to balance the pseudo-labels distribution.

Besides, a non-linear projector head $g\left(\cdot\right)$ is used to map the representation from encoder $f\left(\cdot\right)$ into a low-dimensional embeddings space $z = g \circ f$. The embeddings then are used to compute contrastive loss, which we simplify as:

$$\mathcal{L}_c = \frac{1}{\mu B} \sum_{b=1}^{\mu B} \mathcal{H}\left(q_b^w, q_b^s\right) \tag{3}$$

where $q = \phi\left(norm\left(z\right)\right)$ is the result after the transformation $\phi\left(\cdot\right)$ of CoMatch or SimMatch on the $\mathrm{L}_2$ normalized vector. The momentum embeddings stored in the memory bank and the EMA model are then defined as:

$$z_t \leftarrow m z_{t-1} + (1-m) z_t; \quad \theta_t \leftarrow m \theta_{t-1} + (1-m) \theta_t \tag{4}$$

where $z$ is the embeddings, $\theta$ is the model's parameters, $t$ is the iteration, and $m$ is the momentum parameter. The overall training objective is:

$$\mathcal{L} = \mathcal{L}_s + \lambda_u \mathcal{L}_u + \lambda_c \mathcal{L}_c \tag{5}$$

## 3.2 EPASS

We propose a simple yet effective method to boost the performance of the conventional contrastive-based SSL that maximizes the correctness of the embeddings from different projections by using the ensemble technique.

Unlike conventional methods such as CoMatch and SimMatch, which assume that the learned embeddings from one projector are absolutely correct, we propose using the ensemble embeddings from multiple projectors to mitigate the bias. While there could be diverse options to combine multiple embeddings (e.g., concatenation, summation), we empirically found that simply averaging the selected embeddings works reasonably well and is computationally efficient. As each projector is randomly initialized, it provides a different view of inputs, which benefits the generalization of the model. This intuition is similar to that of multi-view learning. However, since we generate views with multiple projectors instead of creating multiple augmented samples, we introduce far less overhead to the pipeline. The ensemble of multiple projectors helps mitigate the bias in the early stages of training. In the joint-training scheme, the correct learned embeddings help improve the performance of the classification head and vice versa, thus reducing the confirmation bias effect. The embeddings stored in the memory bank by Equation 5 therefore are updated as:

$$z_t \leftarrow m z_{t-1} + (1-m) \bar{z}_t; \quad \bar{z}_t = norm\left(\frac{\sum_{p=1}^{P} z_{t,p}}{P}\right) \tag{6}$$

where $P$ is the number of projectors.

### 3.2.1 Application

**SimMatch:** Using our ensemble embeddings, we re-define instance similarity in SimMatch Zheng et al. (2022) and CoMatch Li et al. (2021) as:

$$\bar{q}_i^w = \frac{exp\left(sim\left(\bar{z}_b^w, \bar{z}_i\right)/T\right)}{\sum_{k=1}^{K} exp\left(sim\left(\bar{z}_b^w, \bar{z}_k\right)/T\right)} \tag{7}$$

where $T$ is the temperature parameter controlling the sharpness of the distribution, $K$ is the number of weakly augmented embeddings, and $i$ represents the $i - th$ instance. Similarly, we can compute $\bar{q}_i^s$ by calculating the similarities between the strongly augmented embeddings $\bar{z}^s$ and $\bar{z}_i$.

$$\bar{q}_i^s = \frac{exp\left(sim\left(\bar{z}_b^s, \bar{z}_i\right)/T\right)}{\sum_{k=1}^K exp\left(sim\left(\bar{z}_b^s, \bar{z}_k\right)/T\right)} \tag{8}$$

The Equation 3 then is rewritten as:

$$\mathcal{L}_c = \frac{1}{\mu B} \sum_{b=1}^{\mu B} \mathcal{H}\left(\bar{q}_b^w, \bar{q}_b^s\right) \tag{9}$$

**CoMatch:** In CoMatch, the embeddings are used to construct a pseudo-label graph that defines the similarity of samples in the label space. Specifically, the instance similarity is also calculated as Equation 7 for weakly augmented samples. Then, a similarity matrix $W^q$ is constructed as:

$$W_{bj}^q = \begin{cases} 1 & \text{if } b = j \\ \bar{q}_b \cdot \bar{q}_j & \text{if } b \neq j \text{ and } \bar{q}_b \cdot \bar{q}_j \geq \tau_c \\ 0 & \text{otherwise} \end{cases} \tag{10}$$

where $\tau_c$ indicates the similarity threshold. Also, an embedding graph $W^z$ is derived as:

$$W_{bj}^z = \begin{cases} \exp\left(\bar{z}_b \cdot \bar{z}_b'/t\right) & \text{if } b = j \\ \exp\left(\bar{z}_b \cdot \bar{z}_j/t\right) & \text{if } b \neq j \end{cases} \tag{11}$$

where $z_b = g \circ f\left(\mathcal{A}_s\left(u_b\right)\right)$ and $z_b' = g \circ f\left(\mathcal{A'}_s\left(u_b\right)\right)$. The Equation 3 then is rewritten as:

$$\mathcal{L}_c = \frac{1}{\mu B} \sum_{b=1}^{\mu B} \mathcal{H}\left(\hat{W}_b^q, \hat{W}_b^z\right) \tag{12}$$

where $\mathcal{H}\left(\hat{W}_b^q, \hat{W}_b^z\right)$ can be decomposed into:

$$\mathcal{H}\left(\hat{W}_b^q, \hat{W}_b^z\right) = -\hat{W}_{bb}^q \log\left(\frac{\exp\left(\bar{z}_b \cdot \bar{z}_b'/T\right)}{\sum_{j=1}^{\mu B} \hat{W}_{bj}^z}\right)$$
$$- \sum_{j=1, j \neq b}^{\mu B} \hat{W}_{bj}^q \log\left(\frac{\exp\left(\bar{z}_b \cdot \bar{z}_j/T\right)}{\sum_{j=1}^{\mu B} \hat{W}_{bj}^z}\right)$$

## 4 Experiments

### 4.1 Implementation Details

We evaluate EPASS on common benchmarks: CIFAR-10/100 Krizhevsky et al. (2009), SVHN Netzer et al. (2011), STL-10 Coates et al. (2011), and ImageNet Deng et al. (2009). We conduct experiments with varying amounts of labeled data, using previous work Sohn et al. (2020); Zhang et al. (2021); Li et al. (2021); Zheng et al. (2022); Xu et al. (2021); Berthelot et al. (2019; 2020); Xie et al. (2020a); Miyato et al. (2019).

For a fair comparison, we train and evaluate all methods using the unified code base USB Wang et al. (2022) with the same backbones and hyperparameters. We use Wide ResNet-28-2 Zagoruyko & Komodakis (2016) for CIFAR-10, Wide ResNet-28-8 for CIFAR-100, Wide ResNet-37-2 Zhou et al. (2020) for STL-10, and ResNet-50 He et al. (2016) for ImageNet. We use SGD with a momentum of 0.9 as an optimizer. The initial learning rate is 0.03 with a cosine learning rate decay schedule of $\eta = \eta_0 \cos\left(\frac{7\pi k}{16K}\right)$, where $\eta_0$ is the initial learning rate and $k(K)$ is the total training step. We set $K = 2^{20}$ for all datasets. During the testing phase,

| Dataset | CIFAR-10 | | | CIFAR-100 | | | SVHN | | | STL-10 | | |
|---|---|---|---|---|---|---|---|---|---|---|---|---|
| Label Amount | 40 | 250 | 4000 | 400 | 2500 | 10000 | 40 | 250 | 1000 | 40 | 250 | 1000 |
| UDA Xie et al. (2020a) | 10.20±5.05 | 5.40±0.28 | 4.27±0.05 | 51.96±1.27 | 29.47±0.52 | 23.59±0.32 | 2.39±0.53 | 1.99±0.02 | 1.91±0.05 | 53.69±4.38 | 28.96±1.02 | 7.25±0.50 |
| MixMatch Berthelot et al. (2019) | 38.84±8.36 | 20.96±2.45 | 10.25±0.01 | 80.58±3.38 | 47.88±0.21 | 33.22±0.06 | 26.61±13.10 | 4.48±0.35 | 5.01±0.12 | 52.32±0.91 | 36.34±0.84 | 25.01±0.43 |
| ReMixMatch Berthelot et al. (2020) | 8.13±0.58 | 6.34±0.22 | 4.65±0.09 | 41.60±1.48 | 25.72±0.07 | **20.04±0.13** | 16.43±13.77 | 5.65±0.35 | 5.36±0.58 | 27.87±3.85 | 11.14±0.52 | 6.44±0.15 |
| FixMatch Sohn et al. (2020) | 12.66±4.49 | 4.95±0.10 | 4.26±0.01 | 45.38±2.07 | 27.71±0.42 | 22.06±0.10 | 3.37±1.01 | 1.97±0.01 | 2.02±0.03 | 38.19±4.76 | 8.64±0.84 | 5.82±0.06 |
| FlexMatch Zhang et al. (2021) | 5.29±0.29 | 4.97±0.07 | 4.24±0.06 | 40.73±1.44 | 26.17±0.18 | 21.75±0.15 | 5.42±2.83 | 8.74±3.32 | 7.90±0.30 | 29.12±5.04 | 9.85±1.35 | 6.08±0.34 |
| Dash Xu et al. (2021) | 9.29±3.28 | 5.16±0.28 | 4.36±0.10 | 47.49±1.05 | 27.47±0.38 | 21.89±0.16 | 5.26±2.02 | 2.01±0.01 | 2.08±0.09 | 42.00±4.94 | 10.50±1.37 | 6.30±0.49 |
| CoMatch Li et al. (2021) | 6.51±1.18 | 5.35±0.14 | 4.27±0.12 | 53.41±2.36 | 29.78±0.11 | 22.11±0.22 | 8.20±5.32 | 2.16±0.04 | 2.01±0.04 | 13.74±4.20 | 7.63±0.94 | 5.71±0.08 |
| SimMatch Zheng et al. (2022) | 5.38±0.01 | 5.36±0.08 | 4.41±0.07 | 39.32±0.72 | 26.21±0.37 | 21.50±0.11 | 7.60±2.11 | 2.48±0.61 | 2.05±0.05 | 16.98±4.24 | 8.27±0.40 | 5.74±0.31 |
| AdaMatch Berthelot et al. (2021) | 5.09±0.21 | 5.13±0.05 | 4.36±0.05 | 38.08±1.35 | 26.66±0.33 | 21.99±0.15 | 6.14±5.35 | 2.13±0.04 | 2.02±0.05 | 19.95±5.17 | 8.59±0.43 | 6.01±0.02 |
| FreeMatch Wang et al. (2023) | **4.90±0.12** | **4.88±0.09** | **4.16±0.06** | 39.52±0.01 | 26.22±0.08 | 21.81±0.17 | 10.43±0.82 | 8.23±3.22 | 7.56±0.25 | 28.50±5.41 | 9.29±1.24 | 5.81±0.32 |
| SoftMatch Chen et al. (2023) | 5.11±0.14 | 4.96±0.09 | 4.27±0.05 | **37.60±0.24** | 26.39±0.38 | 21.86±0.16 | 2.46±0.24 | 2.15±0.07 | 2.09±0.06 | 22.23±3.82 | 9.18±0.68 | 5.79±0.15 |
| **CoMatch + EPASS** | 5.55±0.21 | 5.31±0.13 | 4.23±0.05 | 50.73±0.33 | 29.51±0.16 | 22.16±0.12 | 2.98±0.02 | **1.93±0.05** | **1.85±0.04** | **9.15±3.25** | **6.27±0.03** | **5.40±0.12** |
| **SimMatch + EPASS** | 5.31±0.10 | 5.08±0.05 | 4.37±0.03 | 38.88±0.24 | **25.68±0.33** | 21.32±0.14 | **2.31±0.04** | 2.04±0.02 | 2.02±0.02 | 15.71±2.48 | 8.08±0.26 | 5.58±0.04 |
| Fully-Supervised | 4.62±0.05 | | | 19.30±0.09 | | | 2.13±0.02 | | | None | | |

Table 2: Error rate on CIFAR-10/100, SVHN, and STL-10 datasets on 3 different folds. **Bold** indicates best result and Underline indicates the second best result.

we employ an exponential moving average with a momentum of 0.999 on the training model to perform inference for all algorithms. The batch size for labeled data is 64, with the exception of ImageNet, which has a batch size of 128. The same weight decay value, pre-defined threshold $\tau$, unlabeled batch ratio $\mu$ and loss weights are used for Pseudo-Label Lee et al. (2013), Π model Rasmus et al. (2015), Mean Teacher Tarvainen & Valpola (2017), VAT Miyato et al. (2019), MixMatch Berthelot et al. (2019), ReMixMatch Berthelot et al. (2020), UDA Xie et al. (2020a), FixMatch Sohn et al. (2020), FlexMatch Zhang et al. (2021), CoMatch Li et al. (2021), SimMatch Zheng et al. (2022), AdaMatch Berthelot et al. (2021), and FreeMatch Wang et al. (2023).

We use the same parameters as in Xu et al. (2021); Wang et al. (2022) for Dash method. For other methods, we follow the original settings reported in their studies. In Appendix A, you can find a comprehensive description of the hyperparameters used. To ensure the robustness, we train each algorithm three times with different random seeds. Consistent with Zhang et al. (2021), we report the lowest error rates achieved among all checkpoints.

## 4.2 CIFAR-10/100, STL-10, SVHN

The best error rate of each method is evaluated by averaging the results obtained from three runs with different random seeds. The results are presented in Table 2, where we report the classification error rates on the CIFAR-10/100, STL-10, and SVHN datasets. EPASS is shown to improve the performance of SimMatch and CoMatch significantly on all datasets. For instance, even though EPASS does not achieve state-of-the-art results in CIFAR-10/100, it still boosts the performance of conventional SimMatch and CoMatch. It should be noted that CIFAR-10/100 are small datasets where prior works have already achieved high performance, leaving little room for improvement. Moreover, ReMixMatch performs well on CIFAR-100 (2500) and CIFAR-100 (10000) due to the mixup technique and the self-supervised learning part. Additionally, on the SVHN and STL-10 datasets, SimMatch and CoMatch with EPASS surpass all prior state-of-the-art results by a significant margin, achieving a new state-of-the-art performance. These results demonstrate the effectiveness of EPASS in mitigating bias, particularly on imbalanced datasets such as SVHN and STL-10, where overfitting is a common issue.

## 4.3 ImageNet

EPASS is evaluated on the ImageNet ILSVRC-2012 dataset to demonstrate its effectiveness on large-scale datasets. In order to assess the performance of EPASS, we sample 100k/1%/10% of labeled images in a class-balanced manner, where the number of samples per class is 10, 13, or 128, respectively. The remaining images in each class are left unlabeled. Our experiments are conducted using a fixed random seed, and the results are found to be robust across different runs.

As presented in Table 3, EPASS outperforms the state-of-the-art methods, achieving a top-1 error rate of 39.47%/31.39%/24.70% for SimMatch and a top-1 error rate of 40.24%/32.64%/25.90% for CoMatch,

| Method | Top-1 | Top-5 | Top-1 | Top-5 | Top-1 | Top-5 |
|---|---|---|---|---|---|---|
| | 100k | | 1% | | 10% | |
| FixMatch Sohn et al. (2020) | 43.66 | 21.80 | - | - | 28.50 | 10.90 |
| FlexMatch Zhang et al. (2021) | 41.85 | 19.48 | - | - | - | - |
| CoMatch Li et al. (2021) | 42.17 | 19.64 | 34.00 | 13.60 | 26.30 | 8.60 |
| SimMatch Zheng et al. (2022) | 41.15 | 19.23 | 32.80 | 12.90 | 25.60 | 8.40 |
| FreeMatch Wang et al. (2023) | 40.57 | 18.77 | - | - | - | - |
| SoftMatch Chen et al. (2023) | 40.52 | - | - | - | - | - |
| **CoMatch + EPASS** | 40.24 | 18.40 | 32.64 | 12.71 | 25.90 | 8.48 |
| **SimMatch + EPASS** | **39.47** | **18.24** | **31.39** | **12.41** | **24.70** | **7.44** |

Table 3: ImageNet error rate results. **Bold** indicates best result and Underline indicates the second best result.

respectively. The results clearly demonstrate the effectiveness of EPASS in improving the performance of SSL methods on large-scale datasets like ImageNet.

## 5 Ablation Study

### 5.1 ImageNet convergence speed

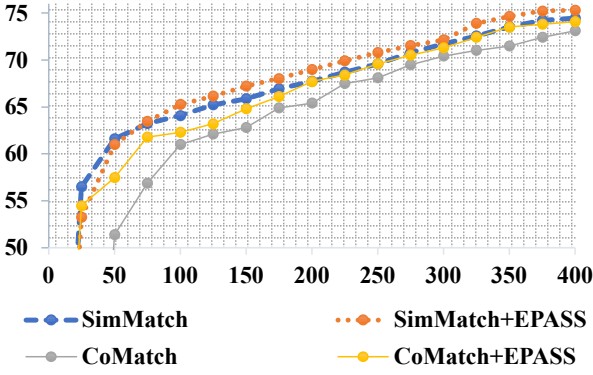

Figure 3: Convergence analysis of SimMatch with and without EPASS.

The convergence speed of the proposed EPASS is extremely noticeable through our extensive experiments. When training on ImageNet, we observe that EPASS achieves over 50% of accuracy in the first few iterations, indicating that the model is able to quickly learn meaningful representations from the unlabeled data. This is likely due to the fact that EPASS encourages the model to focus on the most informative and diverse instances during training, which helps the model learn more quickly and effectively. Additionally, we find that the accuracy of SimMatch and CoMatch with EPASS is consistently increasing with iterations, outperforming conventional SimMatch and CoMatch with the same training epochs. This suggests that the use of EPASS enables the model to continue learning and improving over time, rather than plateauing or becoming overfitted. Overall, these results demonstrate the effectiveness of EPASS in improving the convergence speed and performance of SSL methods.

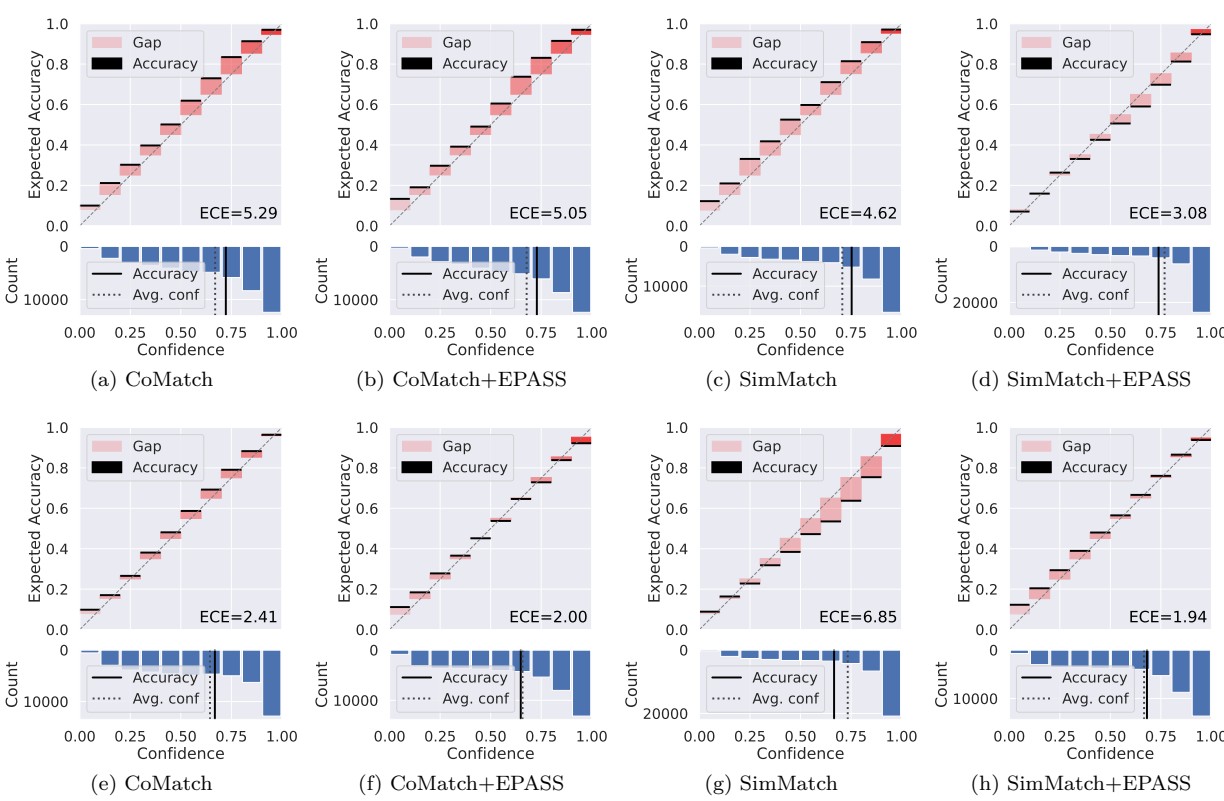

Figure 4: Reliability diagrams (top) and confidence histograms (bottom) for ImageNet dataset. The first row and second row are conducted with 10% and 1% of labels, respectively.

## 5.2 Calibration of SSL

Chen et al. (2022a) propose a method for addressing confirmation bias from the calibration perspective. To evaluate the effectiveness of EPASS in this regard, we measure the calibration of CoMatch and SimMatch on the ImageNet dataset with and without EPASS, using 10% labeled data [1]. Several common calibration indicators, including Expected Calibration Error (ECE), confidence histogram, and reliability diagram, are utilized in this study.

Figure 4 illustrates that when EPASS is used with 10% of labels, the ECE value of the model decreases. Moreover, under the 1% label scheme, CoMatch and SimMatch without EPASS are significantly overconfident and overfitted due to confirmation bias. However, when EPASS is employed, it helps to reduce the ECE by a large margin and also mitigate the overconfidence of the model. Notably, models with EPASS have average accuracy and average confidence that are approximately equal, whereas the average confidence of models without EPASS is usually higher than the accuracy.

It is worth mentioning that since CoMatch does not impose the interaction between semantic and instance similarity like SimMatch, the effect of introducing EPASS to CoMatch for calibration is not as significant as that for SimMatch. Additionally, the model with EPASS becomes underfit and may benefit from additional training.

## 5.3 Number of projectors

This section studies the effectiveness of the proposed projectors ensemble method and how different ensemble strategies affect performance. In this experiment, we study the effect of different numbers of projectors on

---

[1]https://github.com/hollance/reliability-diagrams

performance. The top-1 classification accuracy of the proposed EPASS with different numbers of projectors is shown in Table 4.

| Method \ # projectors | 1 | 2 | **3** | 4 |
|---|---|---|---|---|
| **CoMatch + EPASS** | 73.6 | 73.8 | **74.1** | 73.9 |
| **SimMatch + EPASS** | 74.4 | 74.8 | **75.3** | 75.2 |

Table 4: Top-1 accuracy (%) on ImageNet 10% using different numbers of projectors.

In Table 5, we record the results of different ensemble strategies for EPASS. Overall, averaging the embeddings results in better performance than concatenation and summation.

| Method \ Ensemble strategy | Concatenate | Sum | **Mean** |
|---|---|---|---|
| **CoMatch + EPASS** | 74.0 | 73.9 | **74.1** |
| **SimMatch + EPASS** | 75.1 | 74.8 | **75.3** |

Table 5: Top-1 accuracy (%) on ImageNet 10% using different ensemble strategies.

## 5.4 Imbalanced SSL

| Dataset | CIFAR-10-LT | | CIFAR-100-LT | |
|---|---|---|---|---|
| Imbalance $\lambda$ | $\lambda = 50$ | $\lambda = 150$ | $\lambda = 20$ | $\lambda = 100$ |
| FixMatch Sohn et al. (2020) | $18.5_{\pm 0.48}$ | $31.2_{\pm 1.08}$ | $49.1_{\pm 0.62}$ | $62.5_{\pm 0.36}$ |
| FlexMatch Zhang et al. (2021) | $17.8_{\pm 0.24}$ | $29.5_{\pm 0.47}$ | $48.9_{\pm 0.71}$ | $62.7_{\pm 0.08}$ |
| FreeMatch Wang et al. (2023) | $17.7_{\pm 0.33}$ | $28.8_{\pm 0.64}$ | $48.4_{\pm 0.91}$ | $62.5_{\pm 0.23}$ |
| SoftMatch Chen et al. (2023) | $16.6_{\pm 0.29}$ | $\mathbf{27.4}_{\pm 0.46}$ | $48.1_{\pm 0.55}$ | $61.1_{\pm 0.81}$ |
| CoMatch Li et al. (2021) | $\underline{16.3}_{\pm 0.24}$ | $30.1_{\pm 0.31}$ | $46.2_{\pm 0.41}$ | $60.0_{\pm 0.21}$ |
| SimMatch Zheng et al. (2022) | $20.3_{\pm 0.31}$ | $28.7_{\pm 0.48}$ | $\underline{45.4}_{\pm 0.55}$ | $60.1_{\pm 0.21}$ |
| **CoMatch + EPASS** | $\mathbf{16.1}_{\pm 0.22}$ | $29.6_{\pm 0.41}$ | $45.9_{\pm 0.45}$ | $\underline{59.8}_{\pm 0.01}$ |
| **SimMatch + EPASS** | $18.2_{\pm 0.34}$ | $\underline{28.4}_{\pm 0.43}$ | $\mathbf{45.2}_{\pm 0.51}$ | $\mathbf{59.6}_{\pm 0.11}$ |
| FixMatch + ABC Lee et al. (2021) | $14.0_{\pm 0.22}$ | $22.3_{\pm 1.08}$ | $46.6_{\pm 0.69}$ | $\mathbf{58.3}_{\pm 0.41}$ |
| FlexMatch + ABC Lee et al. (2021) | $14.2_{\pm 0.34}$ | $23.1_{\pm 0.70}$ | $46.2_{\pm 0.47}$ | $58.9_{\pm 0.51}$ |
| FreeMatch + ABC Lee et al. (2021) | $\underline{13.9}_{\pm 0.03}$ | $22.3_{\pm 0.26}$ | $45.6_{\pm 0.76}$ | $58.9_{\pm 0.55}$ |
| CoMatch + ABC Lee et al. (2021) | $14.1_{\pm 0.21}$ | $23.1_{\pm 0.32}$ | $43.0_{\pm 0.52}$ | $59.0_{\pm 0.31}$ |
| SimMatch + ABC Lee et al. (2021) | $14.5_{\pm 0.25}$ | $\underline{20.5}_{\pm 0.21}$ | $43.3_{\pm 0.44}$ | $58.9_{\pm 0.50}$ |
| **CoMatch + EPASS + ABC Lee et al. (2021)** | $14.0_{\pm 0.19}$ | $22.4_{\pm 0.41}$ | $\underline{42.7}_{\pm 0.55}$ | $58.5_{\pm 0.41}$ |
| **SimMatch + EPASS + ABC Lee et al. (2021)** | $\mathbf{13.3}_{\pm 0.09}$ | $\mathbf{20.2}_{\pm 0.26}$ | $\mathbf{42.7}_{\pm 0.41}$ | $58.8_{\pm 0.37}$ |

Table 6: Error rates (%) of imbalanced SSL using 3 different random seeds. **Bold** indicates best result and Underline indicates the second best result.

To provide additional evidence of the effectiveness of EPASS, we assess its performance in the imbalanced semi-supervised learning scenario Lee et al. (2021); Wei et al. (2021); Fan et al. (2022), where both the labeled

| Dataset | CIFAR-100 | | STL-10 | | Euro-SAT | | TissueMNIST | | Semi-Aves |
|---|---|---|---|---|---|---|---|---|---|
| Label Amount | 200 | 400 | 20 | 40 | 20 | 40 | 100 | 500 | 3959 |
| UDA Xie et al. (2020a) | $30.75_{\pm1.03}$ | $19.94_{\pm0.32}$ | $39.22_{\pm2.87}$ | $23.59_{\pm2.97}$ | $11.15_{\pm1.20}$ | $5.99_{\pm0.75}$ | $55.88_{\pm3.26}$ | $51.42_{\pm2.05}$ | $32.55_{\pm0.26}$ |
| MixMatch Berthelot et al. (2019) | $37.43_{\pm0.58}$ | $26.17_{\pm0.24}$ | $48.98_{\pm1.41}$ | $25.56_{\pm3.00}$ | $29.86_{\pm2.89}$ | $16.39_{\pm3.17}$ | $\mathbf{55.73_{\pm2.29}}$ | $\mathbf{49.08_{\pm1.06}}$ | $37.22_{\pm0.15}$ |
| ReMixMatch Berthelot et al. (2020) | $\mathbf{20.85_{\pm1.42}}$ | $\underline{16.80_{\pm0.59}}$ | $30.61_{\pm3.47}$ | $18.33_{\pm1.98}$ | $\underline{4.53_{\pm1.60}}$ | $4.10_{\pm0.37}$ | $59.29_{\pm5.16}$ | $52.92_{\pm3.93}$ | $\mathbf{30.40_{\pm0.33}}$ |
| FixMatch Sohn et al. (2020) | $30.45_{\pm0.65}$ | $19.48_{\pm0.93}$ | $42.06_{\pm3.94}$ | $24.05_{\pm1.79}$ | $12.48_{\pm2.57}$ | $6.41_{\pm1.64}$ | $55.95_{\pm4.06}$ | $50.93_{\pm1.23}$ | $31.74_{\pm0.33}$ |
| FlexMatch Zhang et al. (2021) | $27.08_{\pm0.90}$ | $17.67_{\pm0.66}$ | $37.58_{\pm2.97}$ | $23.40_{\pm1.50}$ | $7.07_{\pm2.32}$ | $5.58_{\pm0.57}$ | $57.23_{\pm2.50}$ | $52.06_{\pm1.78}$ | $33.09_{\pm0.16}$ |
| Dash Xu et al. (2021) | $30.19_{\pm1.34}$ | $18.90_{\pm0.420}$ | $43.34_{\pm1.46}$ | $25.90_{\pm0.35}$ | $9.44_{\pm0.75}$ | $7.00_{\pm1.39}$ | $57.00_{\pm2.81}$ | $50.93_{\pm1.54}$ | $32.56_{\pm0.39}$ |
| CoMatch Li et al. (2021) | $35.68_{\pm0.54}$ | $26.10_{\pm0.09}$ | $\underline{29.70_{\pm1.17}}$ | $21.46_{\pm1.34}$ | $5.25_{\pm0.49}$ | $4.89_{\pm0.86}$ | $57.15_{\pm3.46}$ | $51.83_{\pm0.71}$ | $41.39_{\pm0.16}$ |
| SimMatch Zheng et al. (2022) | $23.26_{\pm1.25}$ | $16.82_{\pm0.40}$ | $34.12_{\pm1.63}$ | $22.97_{\pm2.04}$ | $6.88_{\pm1.77}$ | $5.86_{\pm1.07}$ | $57.91_{\pm4.60}$ | $51.14_{\pm1.83}$ | $34.14_{\pm0.30}$ |
| AdaMatch Berthelot et al. (2021) | $\underline{21.27_{\pm1.04}}$ | $17.01_{\pm0.55}$ | $36.25_{\pm1.89}$ | $23.30_{\pm0.73}$ | $5.70_{\pm0.37}$ | $4.92_{\pm0.87}$ | $57.87_{\pm4.47}$ | $52.28_{\pm0.79}$ | $\underline{31.54_{\pm0.10}}$ |
| **CoMatch + EPASS** | $35.10_{\pm0.55}$ | $25.53_{\pm0.50}$ | $\mathbf{29.56_{\pm2.50}}$ | $\mathbf{21.14_{\pm0.31}}$ | $\mathbf{3.41_{\pm0.24}}$ | $\mathbf{2.91_{\pm0.41}}$ | $56.88_{\pm4.93}$ | $51.06_{\pm1.09}$ | $41.19_{\pm0.43}$ |
| **SimMatch + EPASS** | $22.52_{\pm0.83}$ | $\mathbf{16.78_{\pm0.59}}$ | $30.03_{\pm0.71}$ | $22.65_{\pm1.94}$ | $5.35_{\pm0.81}$ | $\underline{3.81_{\pm0.37}}$ | $57.22_{\pm5.97}$ | $\underline{50.40_{\pm1.44}}$ | $33.83_{\pm0.04}$ |
| Fully-Supervised | $8.90_{\pm0.12}$ | | - | | $0.85_{\pm0.06}$ | | $33.91_{\pm0.03}$ | | - |

Table 7: Error rate on CIFAR-10/100, SVHN, and STL-10 datasets on 3 different folds. **Bold** indicates best result and Underline indicates second best result.

and unlabeled data are imbalanced. Our experiments are conducted on CIFAR-10-LT and CIFAR-100-LT, using varying degrees of class imbalance ratios. For the CIFAR datasets, the imbalance ratio is defined as follows: $\lambda = N_{max}/N_{min}$ where $N_{max}$ is the number of samples on the head (frequent) class and $N_{min}$ the tail (rare). Note that the number of samples for class k is computed as $N_k = N_{max}\lambda^{-\frac{k-1}{C-1}}$, where $C$ is the number of classes. Following Lee et al. (2021); Fan et al. (2022), we set $N_{max} = 1500$ for CIFAR-10 and $N_{max} = 150$ for CIFAR-100, and the number of unlabeled data is twice as many for each class. We use a WRN-28-2 Zagoruyko & Komodakis (2016) as the backbone. We use Adam as the optimizer. The initial learning rate is 0.002 with a cosine learning rate decay schedule as $\eta = \eta_0 \cos\left(\frac{7\pi k}{16K}\right)$, where $\eta_0$ is the initial learning rate, $k(K)$ is the current (total) training step and we set $K = 2.5 \times 10^5$ for all datasets. The batch size of labeled and unlabeled data is 64 and 128, respectively. Weight decay is set as $4e^{-5}$. Each experiment is run on three different data splits, and we report the average of the best error rates.

The results are summarized in Table 6. Compared with other standard SSL methods, EPASS achieves the best performance across all settings. Especially on CIFAR-100 at an imbalance ratio 100, SimMatch with EPASS outperforms the second-best by 0.6%. Moreover, when plugged in the other imbalanced SSL method Lee et al. (2021), EPASS still attains the best performance in most of the settings.

## 5.5 Result using USB

In this section, we evaluate the effectiveness of EPASS within the context of the USB Wang et al. (2022) framework, adhering strictly to the USB settings for CV tasks that utilize pre-trained Vision Transformers (ViT). For a detailed overview of hyperparameters used in these experiments, please refer to Appendix A.

As Table 7 indicates, EPASS improves the performance of SimMatch and CoMatch on all datasets, albeit marginally. These experiments utilize pre-trained ViT models, which provide a strong representation initialization on unlabeled data, leaving little room for improvement when applying SSL methods with this kind of model. Notably, ReMixMatch Berthelot et al. (2020) achieves the highest performance among all SSL algorithms due to its usage of mixup Zhang et al. (2017), Distribution Alignment, and rotation self-supervised loss. However, on CIFAR-100, STL-10, Euro-SAT, and TissueMNIST datasets, EPASS outperforms ReMix-Match.

## 6 Conclusion

Our proposed method, EPASS, enhances the performance and reliability of conventional contrastive joint-training SSL methods. EPASS achieves this by mitigating confirmation bias and embedding bias, which leads to simultaneous performance improvement and reduced overconfidence. EPASS outperforms strong competitors across a variety of SSL benchmarks, especially in the large-scale dataset setting. Additionally, EPASS introduces minimal overhead to the overall pipeline.

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

## A  Hyperparameter setting

We report the detailed hyperparameters setting with a specific model for each dataset in Table 8 and Table 9.

### A.1  Setup for Table 2

For classic CV tasks, we follow the setup from the original papers using USB codebase. The details setup hyperparameters are listed in Table 8.

| Dataset | CIFAR-10 | CIFAR-100 | STL-10 | SVHN | ImageNet |
|---|---|---|---|---|---|
| Model | WRN-28-2 | WRN-28-8 | WRN-37-2 | WRN-28-2 | ResNet-50 |
| Weight Decay | 5e-4 | 1e-3 | 5e-4 | 5e-4 | 3e-4 |
| Labeled Batch Size | | 64 | | | 128 |
| Unlabeled Batch Size | | 448 | | | 128 |
| Learning Rate | | | 0.03 | | |
| SGD Momentum | | | 0.9 | | |
| EMA Momentum | | | 0.999 | | |
| Scheduler | | | $\eta = \eta_0 \cos\left(\frac{7\pi k}{16K}\right)$ | | |
| Weak Augmentation | | | Random Crop, Random Horizontal Flip | | |
| Strong Augmentation | | | RandAugment Cubuk et al. (2020) | | |
| Unsupervised Loss Weight | | | 1 | | |

Table 8: Dataset-wise hyperparameters for classic CV tasks.

### A.2  Setup for Table 7

Pre-trained ViT models Dosovitskiy et al. (2020) are used for CV tasks in USB. For TissueMNIST, CIFAR-100, and Euro-SAT, we use ViT-Tiny and ViT-Small with a patch size of 4 and an image size of 32, while for Semi-Aves, we use ViT-Small with a patch size of 16 and an image size of 224. For STL10, which is a subset of ImageNet, we use unsupervised pre-training MAE He et al. (2022) of ViT-Base with an image size of 96 to prevent cheating.

Following USB CV tasks, we adopt layer-wise learning rate decay as in Liu et al. (2021). The cosine annealing scheduler is used with a total step of 204,800 and warm-up for 5,120 steps. Both labeled and unlabeled batch sizes are set to 16, and other algorithm-related hyper-parameters remain the same as in the original papers.

## B  ImageNet detailed results

Table 10 shows the detailed results from Table 3. EPASS achieves 75.3% of top-1 accuracy with the same training duration ($\sim 400$ epochs) on 10% of labels for SimMatch, and 74.1% of top-1 accuracy for CoMatch. These improvements are also noticeable when EPASS is deployed on 1% of labels, achieving 67.4% and 68.6% top-1 accuracy for CoMatch and SimMatch, respectively.

## C  Precision, Recall, F1 and AUC

We further report precision, recall, F1-score, and AUC (area under curve) results on the CIFAR-10/100, SVHN, and STL-10 datasets. As shown in Table 11 and Table 12, EPASS also has the best performance on

| Dataset | CIFAR-100 | STL-10 | Euro-SAT | TissueMNIST | Semi-Aves |
|---|---|---|---|---|---|
| Image Size | 32 | 96 | 32 | 32 | 224 |
| Model | ViT-S-P4-32 | ViT-B-P16-96 | ViT-S-P4-32 | ViT-T-P4-32 | ViT-S-P16-224 |
| Weight Decay | | | 5e-4 | | |
| Labeled Batch Size | | | 16 | | |
| Unlabeled Batch Size | | | 16 | | |
| Learning Rate | 5e-4 | 1e-4 | 5e-5 | 5e-5 | 1e-3 |
| Layer Decay Rate | 0.5 | 0.95 | 1.0 | 0.95 | 0.65 |
| Scheduler | | | $\eta = \eta_0 \cos\left(\frac{7\pi k}{16K}\right)$ | | |
| Model EMA Momentum | | | 0.0 | | |
| Prediction EMA Momentum | | | 0.999 | | |
| Weak Augmentation | | | Random Crop, Random Horizontal Flip | | |
| Strong Augmentation | | | RandAugment Cubuk et al. (2020) | | |

Table 9: Dataset-wise hyperparameters for USB Wang et al. (2022) CV tasks.

| Self-supervised Pre-training | Method | Epochs | Parameters (train/test) | 1% labels top-1 | top-5 | 10% labels top-1 | top-5 |
|---|---|---|---|---|---|---|---|
| None | FixMatch | $\sim 300$ | 25.6M/25.6M | - | - | 71.5 | 89.1 |
| | CoMatchLi et al. (2021) | $\sim 400$ | 30.0M/25.6M | 66.0 | 86.4 | 73.6 | 91.6 |
| | SimMatchZheng et al. (2022) | $\sim 400$ | 30.0M/25.6M | 67.2 | 87.1 | 74.4 | 91.6 |
| MoCo V2Chen et al. (2020c) | CoMatchLi et al. (2021) | $\sim 1200$ | 30.0M/25.6M | 67.1 | 87.1 | 73.7 | 91.4 |
| MoCo-EMANCai et al. (2021) | FixMatch-EMANCai et al. (2021) | $\sim 1100$ | 30.0M/25.6M | 63.0 | 83.4 | 74.0 | 90.9 |
| **None** | **CoMatch + EPASS** | $\sim 400$ | 30.0M/25.6M | 67.4 | 87.3 | 74.1 | 91.5 |
| **None** | **SimMatch + EPASS** | $\sim 400$ | 30.0M/25.6M | **68.6** | **87.6** | **75.3** | **92.6** |

Table 10: Accuracy results on ImageNet with **1%** and **10%** labeled examples.

precision, recall, F1-score, and AUC on all datasets except CIFAR. Especially on the STL-10 dataset, the improvement from EPASS for CoMatch and SimMatch is very noticeable by a large margin.

| Dataset | CIFAR-10 (40) | | | CIFAR-100 (400) | | |
|---|---|---|---|---|---|---|
| Criteria | Precision | Recall | F1 Score | Precision | Recall | F1 Score |
| UDA | 0.9333 | 0.9311 | 0.9302 | 0.5813 | 0.5484 | 0.5087 |
| FixMatch | 0.9351 | 0.9307 | 0.9297 | 0.5574 | 0.5430 | 0.4946 |
| Dash | 0.8847 | 0.8486 | 0.8210 | 0.5833 | 0.5649 | 0.5215 |
| FlexMatch | 0.9505 | 0.9507 | 0.9505 | 0.6135 | 0.6193 | 0.6107 |
| FreeMatch | **0.9510** | **0.9512** | **0.9510** | **0.6243** | **0.6261** | **0.6137** |
| CoMatch | 0.9441 | 0.9445 | 0.9441 | 0.4543 | 0.3979 | 0.4067 |
| SimMatch | 0.9434 | 0.9438 | 0.9434 | 0.5101 | 0.5133 | 0.5017 |
| **CoMatch + EPASS** | 0.9447 | 0.9450 | 0.9447 | 0.5588 | 0.4927 | 0.4978 |
| **SimMatch + EPASS** | 0.9493 | 0.9494 | 0.9491 | 0.6084 | 0.6061 | 0.6003 |

Table 11: Precision, recall, F1-score and AUC results on CIFAR-10/100.

| Dataset | SVHN (40) | | | STL-10 (40) | | |
|---|---|---|---|---|---|---|
| Criteria | Precision | Recall | F1 Score | Precision | Recall | F1 Score |
| UDA | 0.9781 | 0.9777 | 0.9780 | 0.6385 | 0.5319 | 0.4765 |
| FixMatch | 0.9731 | 0.9706 | 0.9716 | 0.6590 | 0.5830 | 0.5405 |
| Dash | 0.9779 | 0.9777 | 0.9778 | 0.8117 | 0.6020 | 0.5448 |
| FlexMatch | 0.9566 | 0.9691 | 0.9625 | 0.6403 | 0.6755 | 0.6518 |
| FreeMatch | 0.9551 | 0.9665 | 0.9605 | 0.8489 | 0.8439 | 0.8354 |
| CoMatch | 0.9542 | 0.9677 | 0.9605 | - | - | - |
| SimMatch | 0.9718 | 0.9782 | 0.9748 | - | - | - |
| **CoMatch + EPASS** | 0.9647 | 0.9724 | 0.9684 | **0.9100** | **0.9085** | **0.9075** |
| **SimMatch + EPASS** | **0.9782** | **0.9778** | **0.9780** | 0.8026 | 0.8029 | 0.7977 |

Table 12: Precision, recall, F1-score and AUC results on SVHN and STL-10.

## D  List of Data Transformations

We report the detailed augmentations used in our method in Table 13. This list of transformations is similar to the original list used in FixMatch Sohn et al. (2020) and FlexMatch Zhang et al. (2021).

| Transformation | Description | Parameter | Range |
|---|---|---|---|
| Autocontrast | Maximizes the image contrast by setting the darkest (lightest) pixel to black (white). | | |
| Brightness | Adjusts the brightness of the image. $B = 0$ returns a black image, $B = 1$ returns the original image. | $B$ | [0.05, 0.95] |
| Color | Adjusts the color balance of the image like in a TV. $C = 0$ returns a black & white image, $C = 1$ returns the original image. | $C$ | [0.05, 0.95] |
| Contrast | Controls the contrast of the image. A $C = 0$ returns a gray image, $C = 1$ returns the original image. | $C$ | [0.05, 0.95] |
| Equalize | Equalizes the image histogram. | | |
| Identity | Returns the original image. | | |
| Posterize | Reduces each pixel to $B$ bits. | $B$ | [4, 8] |
| Rotate | Rotates the image by $\theta$ degrees. | $\theta$ | [-30, 30] |
| Sharpness | Adjusts the sharpness of the image, where $S = 0$ returns a blurred image, and $S = 1$ returns the original image. | $S$ | [0.05, 0.95] |
| Shear_x | Shears the image along the horizontal axis with rate $R$. | $R$ | [-0.3, 0.3] |
| Shear_y | Shears the image along the vertical axis with rate $R$. | $R$ | [-0.3, 0.3] |
| Solarize | Inverts all pixels above a threshold value of $T$. | $T$ | [0, 1] |
| Translate_x | Translates the image horizontally by ($\lambda \times$image width) pixels. | $\lambda$ | [-0.3, 0.3] |
| Translate_y | Translates the image vertically by ($\lambda \times$image height) pixels. | $\lambda$ | [-0.3, 0.3] |

Table 13: List of transformations used in RandAugment

# E Qualitative Analysis

We present the T-SNE visualization of features on STL-10 test dataset with 40-label split in Figure 5,6. The visualization is using trained models from SimMatch and CoMatch with EPASS.

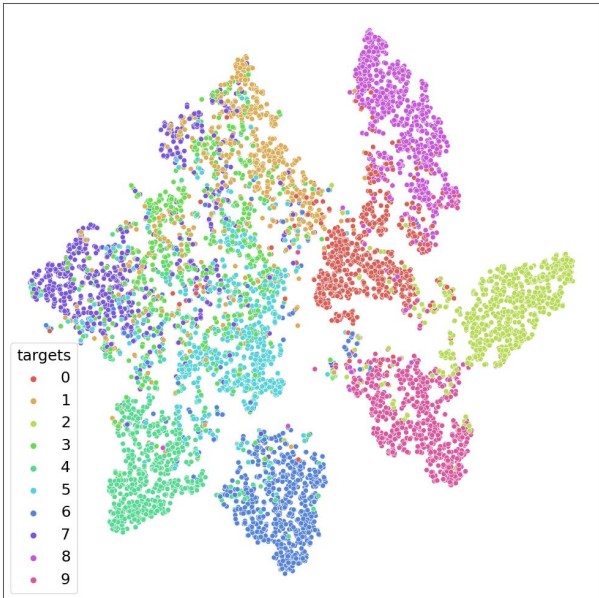

Figure 5: T-SNE visualization of SimMatch + EPASS features on STL-10 dataset with 40-label split.

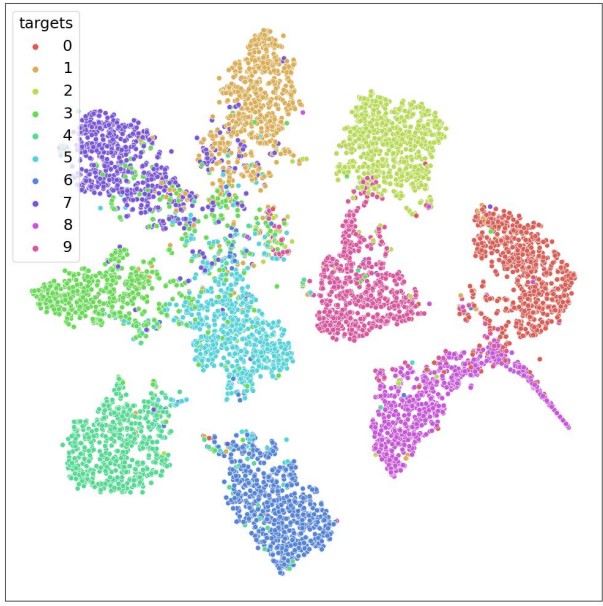

Figure 6: T-SNE visualization of CoMatch + EPASS features on STL-10 dataset with 40-label split.

We also illustrate the T-SNE visualization of features on SVHN test dataset and CIFAR-10 test dataset with 40-label split in Figure 7,8 and Figure 9,10, respectively.

Furthermore, we sketch the T-SNE visualization for the embeddings on those three datasets, as shown in Figures 11, 12, 13, 14, 15, 16, respectively.

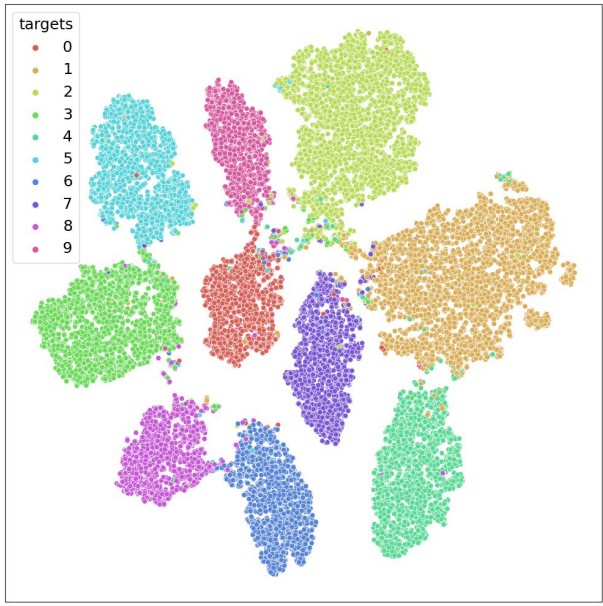

Figure 7: T-SNE visualization of SimMatch + EPASS features on SVHN dataset with 40-label split.

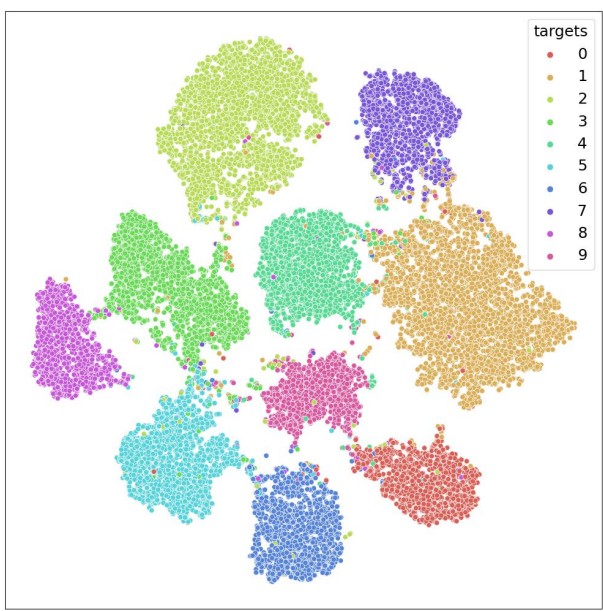

Figure 8: T-SNE visualization of CoMatch + EPASS features on SVHN dataset with 40-label split.

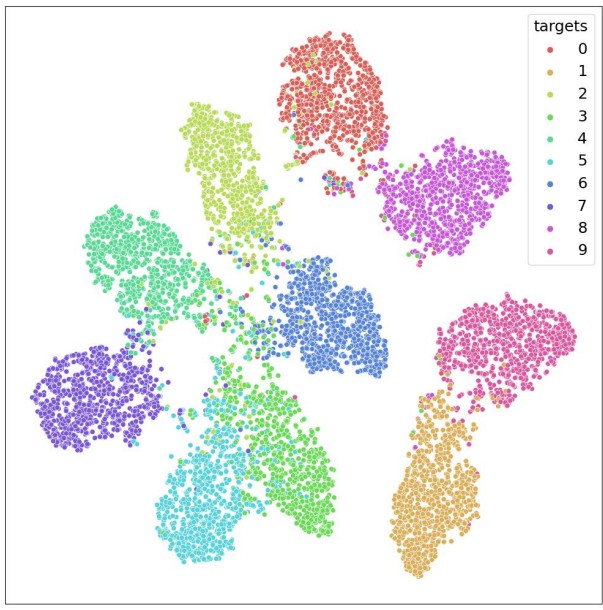

Figure 9: T-SNE visualization of SimMatch + EPASS features on CIFAR-10 dataset with 40-label split.

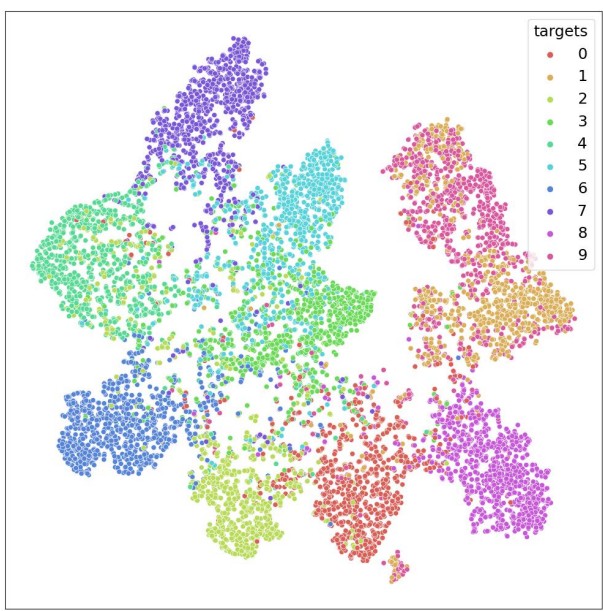

Figure 10: T-SNE visualization of CoMatch + EPASS features on CIFAR-10 dataset with 40-label split.

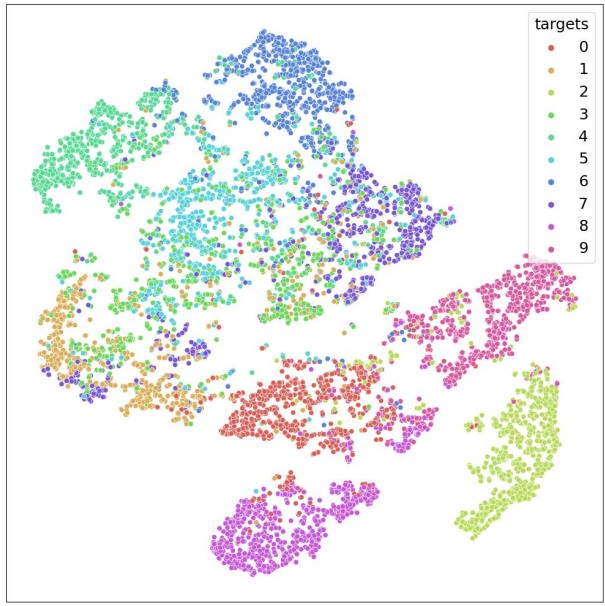

Figure 11: T-SNE visualization of SimMatch + EPASS embeddings on STL-10 dataset with 40-label split.

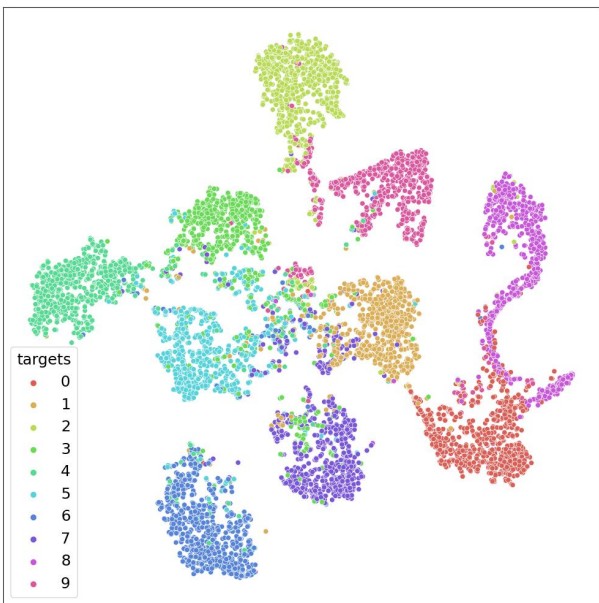

Figure 12: T-SNE visualization of CoMatch + EPASS embeddings on STL-10 dataset with 40-label split.

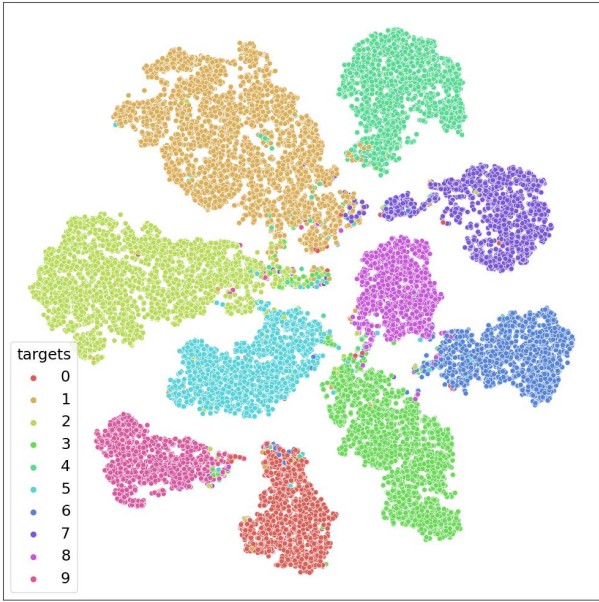

Figure 13: T-SNE visualization of SimMatch + EPASS embeddings on SVHN dataset with 40-label split.

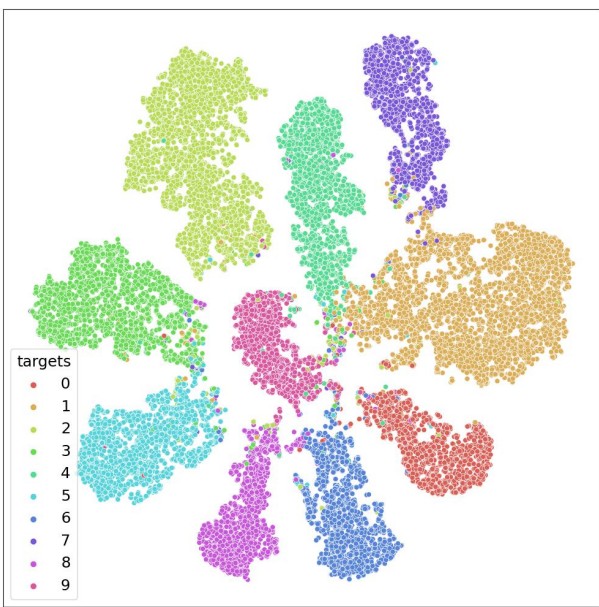

Figure 14: T-SNE visualization of CoMatch + EPASS embeddings on SVHN dataset with 40-label split.

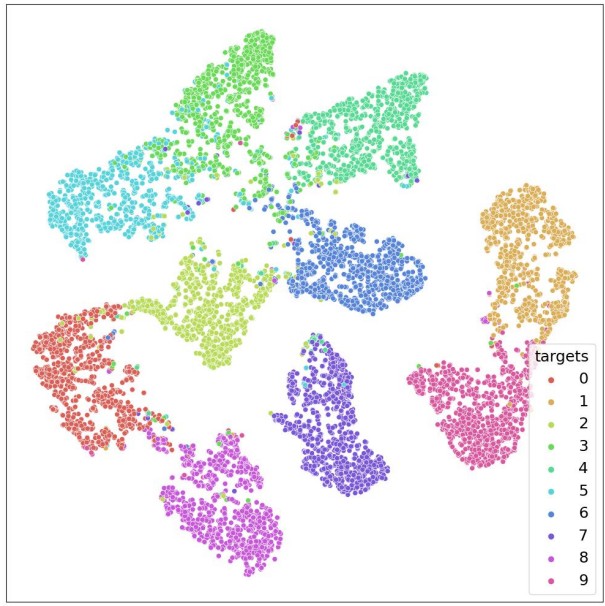

Figure 15: T-SNE visualization of SimMatch + EPASS embeddings on CIFAR-10 dataset with 40-label split.

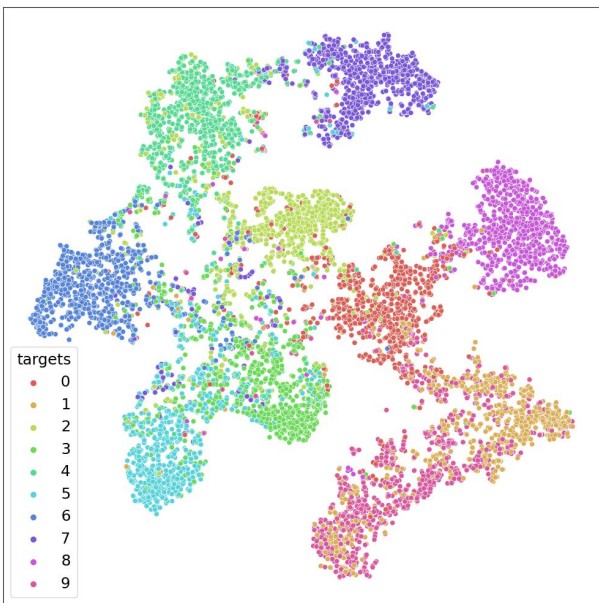

Figure 16: T-SNE visualization of CoMatch + EPASS embeddings on CIFAR-10 dataset with 40-label split.

# F    Algorithm

We apply EPASS to recent state-of-the-art SSL (CoMatch Li et al. (2021) and SimMatch Zheng et al. (2022)) and self-supervised learning (MoCo He et al. (2020)). Applying EPASS to these methods only requires a few lines of code as shown in Algorithm 1.

---

**Algorithm 1:** EPASS

---

**Input:** Encoder $f$, projector $g_k$ and the number of projectors $K$.

**1 for** $b = 1$ **to** $\mu B$ **do**

**2**      Generate prediction distribution as a conventional pipeline by forward propagation.

**3**      **for** $k = 1$ **to** $K$ **do**

**4**          $z_{b,k} = g_k\left(f_b\right)$ // Compute embeddings by different projectors.

**5**      $z_b = norm\left(\frac{\sum_{k=1}^{K} z_{b,k}}{K}\right)$ // Compute the aggregated embeddings.

**6**      Calculate the overall training objective.

**7**      Optimize the model and update the memory bank.

**Output:** The optimized model $f_s$, $h_s$ and $g_{s,k}$.

---

