# OpenReview forum: "Debiasing, calibrating, and improving Semi-supervised Learning performance via simple Ensemble Projector"
_TMLR — Withdrawn by Authors_

### Review · Reviewer_QvYV · 2023-07-30

**Summary Of Contributions:**

This paper introduces a novel method called Ensemble Projectors Aided for Semi-supervised Learning (EPASS) to enhance the performance of existing contrastive joint-training semi-supervised learning frameworks. EPASS is a plug-and-play module that can adapt to various Semi-supervised Learning scenarios. The paper applies this method to CoMatch and SimMatch and conducts comprehensive experiments on CIFAR-10/100, SVHN, STL-10, and ImageNet datasets. Additionally, the paper includes ablation studies to demonstrate how EPASS effectively addresses confirmation bias.

**Audience:**

Yes

**Claims And Evidence:**

No

**Requested Changes:**

Based on the weaknesses identified, I suggested:

1. To strengthen the paper's claims regarding confirmation bias, provide a more detailed analysis of why it occurs in previous methods (e.g., SimMatch, CoMatch) and how EPASS effectively alleviates this issue. Conduct additional ablation studies and analyses to provide stronger evidence of the method's effectiveness in addressing confirmation bias.

2. To ensure a fair comparison with other works, expose more detailed experiment settings and explicitly highlight the differences between EPASS and previous methods. This will help reviewers understand the nuances and make a more informed judgment on the method's performance.

3. In the method section, correct grammar and formula formats to improve clarity. For instance, clarify that yb represents one-hot labels, and U represents a batch of data. Additionally, ensure consistent formatting of equations, such as using the same font for H in equation (2).

4. To enhance readability, streamline the related work section by focusing on the latest or most influential works to cite in the main body. This will make it easier for readers to follow the content and understand the key references. For example, in section 4.1, consider selecting the most relevant and impactful works to cite, rather than including too many citations in a single paragraph.

**Strengths And Weaknesses:**

Strengths:
1. this paper provides a clear and comprehensive explanation of the proposed method, Ensemble Projectors Aided for Semi-supervised Learning (EPASS), which allows readers to grasp the approach easily.
2.  EPASS is a plug-and-play module that can seamlessly integrate into various semi-supervised learning frameworks.

Weaknesses:
1. The method is straightforward, simply add multiple MLP and average the embedding, which the innovation is relatively narrow.
2. The paper's motivation to address confirmation bias is commendable, but section 5.2 lacks strong evidence. To strengthen this aspect, the authors should explicitly explain why previous methods (e.g., SimMatch, CoMatch) suffer from confirmation bias and provide a clear rationale for how EPASS alleviates this issue. Additionally, further justification is needed for choosing ECE as the key metric to address this concern. The experiment settings, which use only 10% and 1% labeled data.
3. The reported improvements in this work are relatively small, especially on CIFAR-10 and CIFAR-100, where the method is not the best setting. To address this concern, providing p-values for the significance of the observed gains would be valuable. This will help convince reviewers of the method's effectiveness.
4. Discrepancies between reported accuracies on STL-10 between the proposed EPASS method and CoMatch raise questions about implementation differences. Clarifying these differences, along with potential sources of variance, will help strengthen the paper's credibility.
5. In section 5.1, the training graph for SimMatch and SimMatch+EPASS appears similar, making it challenging to follow the content and conclusion regarding EPASS improving convergence speed. Consider providing more explicit and detailed analysis or visualizations to demonstrate the claimed improvements clearly.

---

### Review · Reviewer_7um3 · 2023-08-16

**Summary Of Contributions:**

This paper proposes to use multiple projections heads when computing the contrastive loss applied to unlabeled examples during semi-supervised representation learning. The goal of having multiple projections heads seems to be to prevent the model from focusing on only one possible representation; and instead use an ensemble view of the unlabeled samples. Since the ensemble is only at the projections applied to the representation , the approach is more parameter efficient and scalable than using an ensemble of representations/backbone networks. The paper also suggests that this method makes the model more robust. The authors support this with experimental results where their proposed method, titled EPASS, improves error rates over other standard methods like Co-Match and Sim-match

**Audience:**

No

**Broader Impact Concerns:**

I do not see any such concerns.

**Claims And Evidence:**

No

**Requested Changes:**

The paper needs heavy re-writing to be presentable. I am giving a highly non-comprehensive list of suggestions to improve the paper:

Symbol H, and $\mathcal{H}$ have been used interchangeably.

I am confused what $\mathcal{H}(q_b^w, q_b^s)$ means. The notation $L_c$ suggests this is contrastive loss, but then the cross-entropy loss $\mathcal{H}$ is used.

$q$ is used as projections used in contrastive-loss obtained from $\phi$, but $\phi$ isn't defined clearly. It says these are sim-match and co-match transforms. I would suggest a reorganization that individually explains sim-match and co-match, before going onto the epass extension of these methods

The sim-match and co-match descriptions are also poor and needs improvement. As an example, $\bar{q}^s_i$ in Eq 8 is a scalar, but then is used as a vector in Eq 10, and I am not sure in what form in Eq 9.

After Eq 12 $H(q)$ has a definition which uses matrices $W$ but it doesnt clearly define what $\hat{W}^q_b$ means since the matrix is apparently is $W^q_{b,j}$


The authors refer to consistent improvement across methods and network architectures, but the experiment is with only 2 methods, and I do not see training with multiple architectures.

The paper highlighted the lines 'the learned embeddings are correct, regardless of confirmation bias' but gives no explanation of what this is supposed to mean. I would also suggest creating either a synthetic example or some ablation studies which show this 'bias' that the authors are highlighting.


Finally, stepping a bit back, while TMLR is supposed to be concerned more with correctness, I do not see what the community gets from this work. Using ensembles to improve models is common and well studied. And I do not see the performance improvements to be very significant. There needs to be a better evaluation with some good take-away from the work. It might be already there, but it's not clear to me from the writing.



**Strengths And Weaknesses:**

Strengths:

The proposed method is easy to implement and scalable.

Being a model and method agnostic tweak it can also be added on top of various representation learning methods.

The authors have also done experimental analysis with a variety of datasets and included pertinent baseline comparisons.

The proposal is intuitive from a practical standpoint.




Weaknesses:

The paper is quite hard to read, with the introduction feeling very confused.

I spent a lot of time going around trying to interpret the symbols and what exactly is the proposed procedure.

The notation is confusing and adds to the reading difficulty.

Since the paper's main contribution is empirical, the results should be presented in a better fashion along with possible significance. For example, from Table 3 it is not clear how significant is the improvement.

---

### Review · Reviewer_cS4c · 2023-08-17

**Summary Of Contributions:**

This paper considers semi-supervised problems, in particular, those arising in image classification. This paper builds on the contrastive loss, by generating weak augmentations from unlabeled samples. The overall training objective is a combination of supervised objective, unsupervised classification loss, and the contrastive loss from unlabeled data.

The proposed method of this paper, termed EPASS, involves using the ensemble embeddings from multiple projectors to mitigate the bias. The ensemble embeddings can also be applied on top of SimMatch and CoMatch.

Lastly, experiments on CIFAR, SVHN, STL, and ImageNet datasets were conducted, comparing the proposed EPASS framework to existing semi-supervised learning approaches.

**Audience:**

Yes

**Broader Impact Concerns:**

This paper investigates the algorithmic properties of SSL; thus, concerns about its ethical implications are limited.

**Claims And Evidence:**

No

**Requested Changes:**

- The claims regarding debiasing and calibrating SSL should be stated more explicitly. For instance, this could be achieved by reorganizing the experimental results and adjusting the presentations.

- The method description in Section 3 needs to be expanded and needs to include more details about the motivations of the design.


**Strengths And Weaknesses:**

S1) A simple method that should be easy to use for other researchers interested in SSL.

S2) The paper is nicely structured and is thus easy to follow.

S3) Extensive ablation studies help justify the claims made in this paper.

W1) I'm not sure if the claims concerning debiasing (in the title) have been articulated sufficiently in detail. Similarly, there are claims concerning calibration, but this is only stated in Section 5.2 in the ablation studies. These two claims should be stated more explicitly together with the corresponding results. The current paper did not provide sufficient details to connect these claims together.

W2) Similarly, I find that the method description lacks details to fully understand the proposed method; For instance, it is not clearly defined which "norm" is equation (6) using. Are labeled samples used only in $\mathcal{L}_s$?

W3) Lack of rigorous analysis of the proposed method; Would it be possible to justify why the proposed method is expected to work well in practice? Even just providing some illustrative examples could be helpful.

---

### Note · Authors · 2023-08-29

I have read and agree with the venue's withdrawal policy on behalf of myself and my co-authors.